# Combination of Marker-Assisted Backcross Selection of *Yr59* and Phenotypic Selection to Improve Stripe Rust Resistance and Agronomic Performance in Four Elite Wheat Cultivars

**Min Zhang** [1,†]**, Taohong Fang** [1,†]**, Xinli Zhou** [1,*]**, Xianming Chen** [2]**, Xin Li** [1]**, Junyan Feng** [3]**, Suizhuang Yang** [1] **and Zhensheng Kang** [4]

1   Wheat Research Institute, School of Life Sciences and Engineering, Southwest University of Science and Technology, Mianyang 621000, China; maggie_zm2893@sina.com (M.Z.); zrd_479@sina.com (T.F.); sadoneli@gmail.com (X.L.); yangszh@126.com (S.Y.)
2   USDA-ARS, Wheat Health, Genetics, and Quality Research Unit, Department of Plant Pathology, Washington State University, Pullman, WA 99164-6430, USA; xianming@wsu.edu
3   Biotechnology and Nuclear Technology Research Institute, Sichuan Academy of Agricultural Sciences, Chengdu 610066, China; junyanfeng@live.cn
4   State Key Laboratory of Crop Stress Biology in Arid Areas, College of Plant Protection, Northwest A&F University, Xianyang 712100, China; kangzs@nwsuaf.edu.cn
*   Correspondence: eli6951@sina.com
†   These authors contributed equally to this work.

**Abstract:** In this study, we successfully introgressed and validated *Yr59* into four elite wheat cultivars, Jimai 22, Chuanmai 42, Zhengmai 9023 and Xinmai 26 through marker-assisted backcross selection. Used as female parents, these four cultivars were crossed with wheat line PI 660061 (*Yr59*). After two backcrosses and marker-assisted selection, the progenies were selfed and advanced to the $BC_2F_4$ generation. A total of 123 $BC_2F_4$ lines were selected based on agronomic traits and stripe rust resistance, and their $BC_2F_5$ and $BC_2F_6$ progenies were further evaluated for stripe rust resistance and agronomic traits. Seven markers linked with relevant genes, including *Xbarc32*, *Xwgp5175*, *Xwmc557* and *Xcfa2040* linked with *Yr59*; *Xwmc658* with *YrJ22*; *WE173* and *Xbarc181* with *Yr26*, were used to genotype the breeding lines. A total of 109 introgression lines with positive markers for *Yr59* were identified for further stripe rust and agronomic trait evaluation. Finally, 16 lines had higher levels resistance to stripe rust, and similar or superior agronomic traits compared to their parents were obtained. These lines can be released as new cultivars for various regions after regional tests and also can be used as resistance stocks for regional breeding programs to develop new cultivars with adequate and durable resistance to stripe rust.

**Keywords:** wheat stripe rust; high-temperature adult-plant (HTAP) resistance; marker-assisted selection; agronomic traits

## 1. Introduction

Wheat is one of the most widely cultivated cereal crops in the world and a staple food for approximately 40% of the world's populations, providing over 20% of the calories, 25% of the protein and nearly 55% of the carbohydrate intake of humans [1]. The growing human population is posing a concomitant upsurge in foodstuff demand. Therefore, wheat cultivars with high yield and high quality are crucial for global food safety. Certain biotic and abiotic stresses limit the realization of the full genetic potential of modern wheat cultivars, of which stripe rust, caused by *Puccinia striiformis* Westend. f. sp. *tritici Erikss*. Pst, is one of the most important pathogens threatening wheat production [2,3]. It is reported that the grain yield and quality loss of resistant wheat varieties were less than that of the susceptible under wheat stripe rust infection environment [4]. Fungicides can be used to

reduce yield losses caused by stripe rust, but the use of fungicides adds cost to production and may be harmful to the users and the environment. The best approach of stripe rust control is breeding stripe rust-resistant cultivars.

Developing a wheat cultivar through a conventional approach usually takes 7 to 12 years. Marker-assisted selection (MAS) can accelerate the breeding process [5,6]. The MAS breeding method has been wildly used in the improvement of many crops, for instance, barley [7], wheat [8,9], maize [10], peanut [11,12], rice [13], tea [14]. Different types of markers can be used in MAS. Wheat scientists are still using simple sequence repeat (SSR) markers in molecular mapping and MAS, for reproducibility, multi-allelic nature, co-dominant inheritance, good genome coverage and robust amplification and availability of SSR for many specific traits [15]. For example, Vishwakarma et al. [16] used SSR markers to improve grain protein content and grain weight in Indian bread wheat. Yaniv et al. [17] used a set of polymorphic SSR markers to introgress *Yr15* into wheat lines for improving stripe rust resistance. Mallick et al. [18] used markers linked to *Lr19/Sr25* and *Yr10* to combine multiple rust resistances in wheat. Randhawa et al. [19] used SSR markers in transferring stripe and stem rust resistance genes *Yr51*, *Yr57*, *Sr22*, *Sr26* and *Sr50* into four wheat cultivars. In addition, technologies such as high-throughput arrays and next-generation DNA sequencing (NGS) offer other choices of markers and single nucleotide polymorphisms (SNPs), for instance, to study the distribution of chromosomal segments, discovery of new genes and MAS [20,21]. With the characteristic of high-abundance, high-throughput analysis and relatively low genotyping error rates and especially the ability to be transformed to another form of markers in an excellent conversion rate range from 50% to 97%, SNP markers may open the innovation of research gate [22–24]. Kompetitive Allele Specific PCR (KASP) genotyping, an application of SNP marker, has been employed in wheat mapping and breeding. Mu et al. [25] developed KASP markers for stripe rust resistance quantitative trait locus (QTL) *QYrcen.nwafu-7BL*, and Neelam et al. [26] developed KASP markers for leaf rust resistance gene *Lr21*. Using SSR and KASP markers, Qie et al. [27] pyramided tightly linked stripe rust resistance genes *Yr15* and *Yr64* into new wheat germplasm lines. Kaur et al. [28] developed KASP assays for improving wheat resistance or tolerance to various biotic and abiotic stresses and agronomic, physiological and quality traits.

There are two types of resistance to stripe rust: all-stage resistance (ASR, also called seedling resistance) and adult-plant resistance (APR, including high-temperature adult-plant (HTAP) resistance) [29,30]. All-stage resistance is expressed at all growth stages and can be easily detected at the seedling stage [29,31,32]. It is generally race-specific, qualitatively inheritance and readily overcome by newly virulent races. Wheat cultivars containing this type of resistance alone will become susceptible within a few years of release [31]. New virulent races are selected by directional selection pressure from growing cultivars with race-specific resistance [33,34]. By contrast, adult-plant resistance or HTAP resistance is expressed at post-seedling growth stage, is non-race-specific and contains durable resistance [29,31,35,36]. Many genes or QTL for HTAP resistance to stripe rust have been mapped, and markers linked to these genes have been developed. For example, *Yr59*, a HTAP gene reported by Zhou et al. [36], was located on long arm of chromosome 7B and flanked by markers *Xwgp5175* and *Xbarc32* about 2.1 cM in distance [36]. The constant evolvement of virulent Pst races continues circumvents race-specific resistance in wheat cultivars, resulting significant yield losses. It is urgent to introgress APR genes into elite cultivars to combat stripe rust.

So far, 83 officially named genes and at least 300 QTL have been reported, and useful molecular markers have been developed for some of them [37,38]. Over 60 of the stripe rust resistance genes or QTL have been intentionally used in wheat breeding [39], for instance, *Yr5* and *Yr15* combined in several wheat cultivars (reviewed by Wang and Chen 2017), *Yr15* and *Yr64* by Qie et al. [27], *Yr15* by Kaur et al. [40], *Yr26* by Zheng et al. [41], *Yr48* by Yang et al. [42], and *QYr.nafu-2BL* and *QYr.nafu-3BS* by Hu et al. [43].

The present study was conducted to introgress HTAP resistance gene *Yr59* from PI 660061 into four main Chinese elite cultivars from different wheat production regions of China through MAS. Progeny lines were also selected through evaluation of several agronomic traits. The selected lines were demonstrated to have combinations of improved stripe rust resistance and desirable agronomic traits. These lines can be used for further yield and adaptation tests for releasing as new cultivars to be grown in various wheat production regions and/or use as genetic stocks for developing new wheat cultivars with adequate levels of durable resistance to stripe rust.

## 2. Materials and Methods

### 2.1. Plant Material

Four elite wheat cultivars, Chuanmai 42 (CM42), Jimai 22 (JM22), Zhengmai 9023 (ZM9023) and Xinmai 26 (XM26), were used as recurrent parents in this study. These cultivars were selected because they have excellent agronomic traits and have been widely planted in their corresponding regions covering the major wheat growing regions in China [44–47]. CM42, a spring wheat cultivar developed from the cross of Syn-CD768/SW3243//C6415 by the Crop Research Institute of Sichuan Academy of Agricultural Sciences, has high yield potential, disease resistance and good quality and has been widely grown in Sichuan and other provinces in southwestern China. It has *Yr26* [48,49], a stripe rust resistance gene recently becoming ineffective to new races virulent. JM22, a facultative winter cultivar developed by Crop Research Institute, Shandong Academy of Agricultural Sciences and widely grown in Hebei and other provinces in the northern wheat growing region of China, contains *YrJ22* and *PmJM22*, providing only a moderate level of stripe rust resistance [50,51]. ZM9023, a weak spring cultivar developed by Wheat Research Institute of Henan Academy of Agricultural Sciences, has high salt tolerance, superior gluten quality and Fusarium head blight resistance (*QFhb.7D*) [52,53]; it was extensively planted in Henan and other provinces. XM26, developed by Xinxiang Henan Academy of Agricultural Sciences, is a facultative winter wheat variety widely grown in Henan provinces in the Huang Huai Hai wheat growing region. All these four wheat cultivars have high capacity to tolerate local biotic and abiotic stresses but either lost their resistance or do not have adequate resistance to stripe rust under the prevalence of CYR34, a new virulent race found in 2008 in Sichuan [54,55]. PI 178759 is a spring Iraq landrace with *Yr59* for non-race-specific HTAP resistance to stripe rust [36]. PI 660061, a wheat germplasm line transferred from PI 178759, has excellent resistance to stripe rust and relatively better agronomic traits compared to PI 178759 [56] and was used as the donor parent of *Yr59* in crosses with the four Chinese cultivars.

The four crosses (CM42/PI660061, JM22/PI660061, ZM9023/PI660061 and XM26/PI660061) were made in 2014. Two-round backcrosses were completed during 2015-2016, and the $BC_1F_1$ and $BC_2F_1$ plants were selected for the presence of molecular markers for *Yr59*. The selected $BC_2F_1$ plants were selfed and advanced to the $BC_2F_6$ generation from 2016 to 2021 in the fields. Seeds of $BC_2F_2$ to $BC_2F_3$ were bulk-harvested for each cross to keep all possible genotypes. Plants with good stripe rust resistance and agronomic traits were visually selected from the $BC_2F_4$ generation to produce individual $BC_2F_5$ lines. The $BC_2F_5$ and $BC_2F_6$ lines were tested with markers for *Yr59* to selected homozygous resistant lines. The tests before 2017 were conducted in Yangling (34.292N, 108.077E) of Shanxi province, and the tests during 2018–2021 were mainly conducted in Mianyang (31.682N, 104.663E) of Sichuan province.

### 2.2. Phenotyping for Stripe Rust Reaction and Agronomic Traits in the Fields

Phenotypic evaluation and selection were performed on the $BC_2F_4$, $BC_2F_5$ and $BC_2F_6$ generations. A total of 123 individual plants were selected from the BC2F4 generation of all crosses to harvest $BC_2F_5$ seeds based on medium plant height, high stripe rust resistance and good yield-related traits. The $BC_2F_5$ seeds were planted, and their plants were evaluated for stripe rust resistance and agronomic traits in Mianyang during the 2019–2020

growth season, and the $BC_2F_6$ line were planted and evaluated in the experimental fields of both Mianyang and Yangling during the 2020-2021 season. The field experiments were conducted as in a randomized complete block design with three replications. Approximately 60–80 seeds each line were planted in a row of 2 m with 25 cm between rows. The parents and Mingxian 169 (MX169), a stripe rust-susceptible cultivar used as control, were planted after every 20 rows and around the field. Mianyang, where Pst can over-winter and over-summer, is one of the major stripe rust epidemic areas in China, and stripe rust occurs naturally with no need of artificial inoculation [57]. In Yangling, where severe stripe rust occurs less frequently than Mianyang, the field was inoculated with a mix of urediniospores of Pst races CYR32, CYR33 and CYR34 when flag leaves were emerging.

Stripe rust infection type (IT) was assessed based on the 0–9 scale [58], and disease severity (DS) was recorded as the percentage of infected leaf area for each line using a modified scale from Peterson [59]. Both IT and DS data were collected when the susceptible control MX169 exhibited nearly 80% severities around the flowering stage and again after a week. The mean IT and DS scores of the plants and between the two records were calculated and used to rate each line. IT 0–3, 4–6 and 7–9 were considered resistant, intermediate and susceptible, respectively. The DS data were also considered in the line selections.

Desirable agronomic traits such as PH (plant height), NS (number of spikes), SL (spike length), GNS (grain number per spike) and TGW (thousand-grain weight) were determined to select lines. The PH was measured from the ground to the top of the spike except awn after the milk stage. The NS, SL and GNS of each plant were counted after maturity or harvest. Two hundred seeds were randomly sampled three times and weighed for each line, and TGW were calculated as the mean of the three weighs.

### 2.3. Molecular Marker Genotyping

Molecular markers linked to *Yr59*, namely *Xbarc32*, *Xwgp5175*, *Xwmc557* and *Xcfa2040*, were used to select plants carrying the *Yr59* from a previous study [36]. The genetic distances between *Xbarc32* and *Xwgp5175*, *Xwmc557* and *Xcfa2040* are both 2.1 cM, and *Yr59* is located in the *Xbarc32* and *Xwgp5175* interval. *Xwgp5175* linked to *Xwmc557* with genetic distance of 2.2 cM. Of these markers, *Xbarc32*, which is closest to the resistance allele, was used to select $F_1$, $BC_1F_1$ and $BC_2F_1$ plants containing *Yr59*. Markers *Xbarc181* and *WE173*, which are linked to *Yr26* with genetic distances of 6.7 cM and 1.4 cM in CM42, respectively [49], were used to detect $BC_2F_5$ lines containing *Yr26* in the cross of CM42///CM42//CM42/PI660061. *Xwmc658*, which is linked to *YrJ22* with genetic distance of 1.0 cM in JM22 [50], was used to detect *YrJ22* in $BC_2F_5$ lines of JM22///JM22//JM22/PI660061. SSR markers *Xbarc32* and *Xwmc557* were tested by polyacrylamide gel electrophoresis (PAGE) at Southwest University of Science and Technology. Markers *Xwgp5175*, *Xcfa2040*, *Xbarc181*, *WE173* and *Xwmc658* were detected using Fragment Analyzer system at Biotechnology and Nuclear Technology Research Institute, Sichuan Academy of Agricultural Sciences. The sequences of these primers and annealing temperatures are listed in Table 1.

**Table 1.** Sequence and amplification information for simple sequence repeat (SSR) markers linked to *Yr59*, *Yr26* and *YrJ22*.

| Marker | Primer Sequence | Tm (°C) | Linked to *Yr* Gene | References |
|---|---|---|---|---|
| *Xbarc32* | GCGTGAATCCGGAAACCCAATCTGTG<br>TGGAGAACCTTCGCATTGTGTCATTA | 60.5 | *Yr59* | [36,60] |
| *Xwgp5175* | GGAGGCTTAGGGAAG<br>TGGTAGGTCCTTGTA | 49.0 | *Yr59* | [36,60] |
| *Xwmc557* | GGTGCTTGTTCATACGGGCT<br>AGGTCCTCGATCCGCTCAT | 57.6 | *Yr59* | [36,61] |
| *Xcfa2040* | TCAAATGATTTCAGGTAACCACTA<br>TTCCTGATCCCACCAAACAT | 52.2 | *Yr59* | [36,60] |
| *Xwmc658* | CTCATCGTCCTCCTCCACTTTG<br>GCCATCCGTTGACTTGAGGTTA | 57.6 | *YrJ22* | [50,60] |
| *WE173* | GGGACAAGGGGAGTTGAAGC<br>GAGAGTTCCAAGCAGAACAC | 58.0 | *Yr26* | [49,60] |
| *Xbarc181* | CGCTGGAGGGGGTAAGTCATCAC<br>CGCAAATCAAGAACACGGGAGAAAGAA | 58.0 | *Yr26* | [49,60] |

Genomic DNA was extracted from fresh seedling leaves using the Cetyltrimethylammonium bromide (CTAB) method with slight modifications [62]. DNA quality and quantity were checked using Nanodrop ND-1000 spectrophotometer (Nanodrop Technologies, Wilmington, NC, USA), and the stock DNA was diluted to 50 ng/μL for use as template in polymerase chain reaction (PCR).

The PCR reaction for each DNA sample was performed in a 10 μL mixture containing 2 μL (50 ng/μL) DNA template, 1 μL 10× PCR buffer (containing mg2+), 0.8 μL 2.5 mM of dNTP, 1 μL (2 μM) of each primer solution, 0.2 μL Taq DNA polymerase solution (2.5 unit/μL) and 4 μL sterilized double distilled water (ddH2O). The PCR amplification was performed as follows: pre-denaturation at 95 °C for 3 min; followed by 40 cycles of 95.0 °C for 30 s for denaturation, 45 °C or 65 °C for 30 s for annealing depending upon primers, and 72 °C for 30 s for extension; and final 72 °C for 5 min for incubation. A mixture of 4 μL of the amplification products and 6 μL denaturation buffer were loaded in a 6% polyacrylamide gel (Beijing Solarbio Science & Technology Co., Ltd. Beijing, China) [63] or 5 μL of the amplification products mixed with 19 μL TE buffer were separated using Fragment Analyzer (Agilent Technology Co., Ltd. Santa Clara, California, USA). The fluorescence signals of PCR products and genescan-500 molecular weight standard were automatically recorded by the gene analyzer.

*2.4. Statistical Analysis*

Phenotypic data of stripe rust and agronomic traits were analyzed using software GraphPad Prism Version 8.0.2. Histograms were generated using the Grouped function. The unpaired *t*-test was used to compare the means of the traits among wheat lines. A *p*-value of <0.05 was considered significant.

**3. Results**

*3.1. Marker-Assisted Backcrossing (MABC)*

Crossing, backcrossing and selfing of each cross were conducted as in the scheme illustrated in Figure 1. The numbers of selected plants in each generation are given in Table 2. Forty-three "$F_1$" seeds were harvested from the four crosses. Thirty three true $F_1$ plants were identified to contain positive *Xbarc32*, consisting of eight from CM42/PI660061, nine from JM22/PI660061, seven from ZM9023/PI660061 and nine from XM26/PI660061. These true $F_1$ plants were used as male parents to make the first-round backcross with their recurrent parents CM42, JM22, ZM9023 or XM26. About 50 $BC_1F_1$ seeds were harvested from each cross and planted in the greenhouse. DNA extraction and marker testing were performed in the same with $F_1$ plants. After testing with *Xbarc32*, 23 $BC_1F_1$ plants, including 6 from

13 CM42//CM42/PI660061, 7 $BC_1F_1$ plants from 15 JM22//JM22/PI660061, 4 $BC_1F_1$ plants from 10 ZM9023//ZM9023/PI660061 and 6 $BC_1F_1$ plants from 12 XM26//XM26/PI660061, were found heterozygous for the marker locus. These $BC_1F_1$ plants were used to make the second-round backcross. A total of 98 $BC_2F_1$ seeds were obtained and sown in the field in Yangling in 2015 and selfed. After testing with *Xbarc32*, 49 $BC_2F_1$ plants were found to have the marker, including 12 from CM42///CM42//CM42/PI660061, 16 from JM22///JM22//JM22/PI660061, 10 from ZM9023///ZM9023//ZM9023/PI660061 and 11 from XM26///XM26//XM26/PI660061. $BC_2F_2$ seeds were harvested from the 49 $BC_2F_1$ plants, planted in fields, selfed and advanced to the $BC_2F_6$ generation. In the $BC_2F_4$ generation, 123 plants were selected based on their stripe rust reaction and agronomic traits, and the seeds were individually harvested to form 123 $BC_2F_5$ lines, consisting of 28 from CM42///CM42//CM42/PI660061, 43 from JM22///JM22//JM22/PI660061, 19 from ZM9023///ZM9023//ZM9023/PI660061, and 33 from XM26///XM26//XM26/PI660061. The selected $BC_2F_4$ plants had intermediate to resistant reactions to stripe rust, 80-110 cm plant height, and kernel-full spikes. After the selected $BC_2F_5$ lines with two markers (*Xbarc32*, *Xwmc557*) and their $BC_2F_6$ lines with four markers (*Xbarc32*, *Xwgp5175*, *Xwmc557* and *Xcfa2040*), which produced generally consistent results with the disease ratings in the fields of the two locations and two years, 109 lines were selected as resistant to stripe rust with *Yr59* and desirable agronomic traits. The 109 lines included 28 from CM42///CM42//CM42/PI660061, 39 from JM22///JM22//JM22/PI660061, 16 from ZM9023///ZM9023//ZM9023/PI660061 and 26 from XM26///XM26//XM26/PI660061. In addition, markers *Xbarc181* and *WE173* linked to *Yr26* were used to detect the genes in the selected lines from CM42///CM42//CM42/PI660061, and marker *WMC658*, linked to *YrJ22*, was used to detect the gene in the selected lines from XM26///XM26//XM26/PI660061, resulting 15 lines from CM42///CM42//CM42/PI660061 with *Yr26* and 27 lines with *YrJ22* from the crosses, respectively. Among the 109 lines, 15 had both *Yr59* and *Yr26* and 23 had both *Yr59* and *YrJ22* (Table 2).

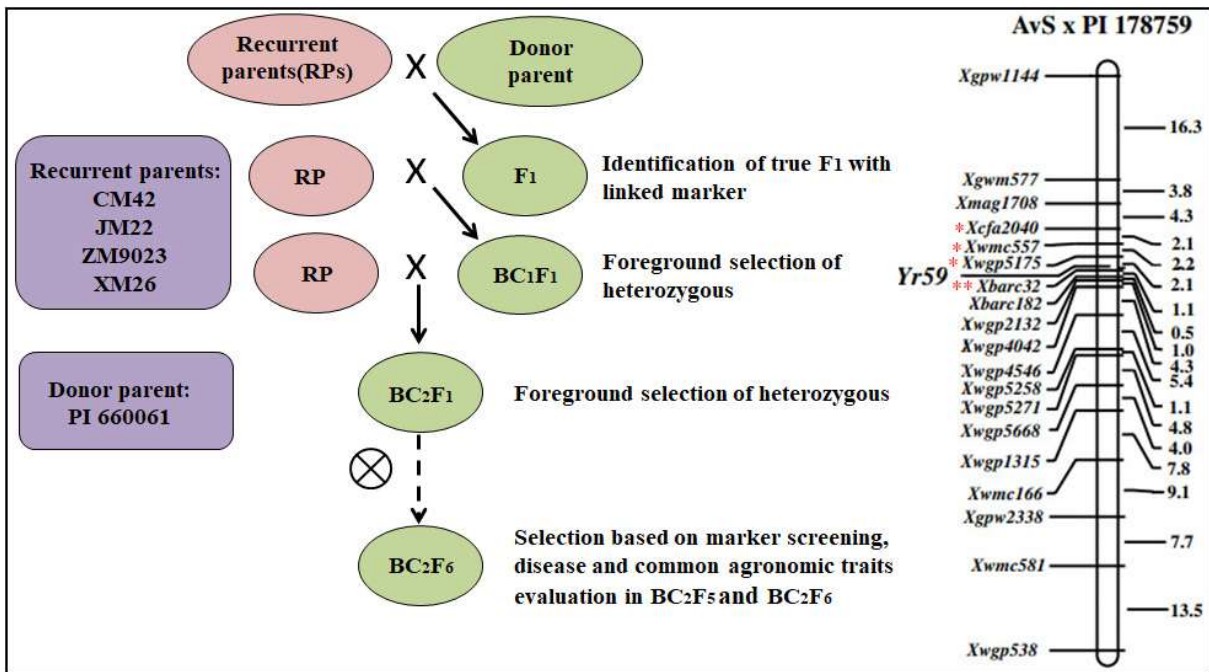

**Figure 1.** The scheme of introgressing stripe rust resistance gene *Yr59* into four elite wheat cultivars through marker-assisted backcrossing (MABC). The *Yr59* linkage map was adopted from Zhou et al. (2014). The marker with two asterisks (**) was used in genotyping $F_1$, $BC_1F_1$ and $BC_2F_1$ plants, and the markers with one (*) and two asterisks (**) were used to screen the $BC_2F_5$ and $BC_2F_6$ lines.

**Table 2.** Numbers of progeny plants (for $F_1$, $BC_1F_1$ and $BC_2F_1$) or lines ($BC_2F_5$ and $BC_2F_6$) tested and positive for *Yr59*-linked SSR markers in different generations derived from wheat crosses of four elite cultivars (CM42, JM22, ZM9023 and XM26) with *Yr59* donor PI 660061). The $F_1$, $BC_1F_1$, $BC_2F_1$ generations were screened with *Xbarc32*, and $BC_2F_5$ and $BC_2F_6$ were screened with *Xbarc32*, *Xwgp5175*, *Xwmc557* and *Xcfa2040*.

| Generations | Number of Plants/Lines | | | | | | | | | |
| | CM42/PI660061 | | JM22/PI660061 | | ZM9023/PI660061 | | XM26/PI660061 | | Total | |
| | Tested | Positive | Tested | Positive | Tested | Positive | Tested | Positive | Tested | Positive |
|---|---|---|---|---|---|---|---|---|---|---|
| $F_1$ | 10 | 8 | 13 | 9 | 9 | 7 | 11 | 9 | 43 | 33 |
| $BC_1F_1$ | 13 | 6 | 15 | 7 | 10 | 4 | 12 | 6 | 50 | 23 |
| $BC_2F_1$ | 24 | 12 | 31 | 16 | 20 | 10 | 23 | 11 | 98 | 49 |
| $BC_2F_5$ | 28 | 28 | 43 | 39 | 19 | 16 | 33 | 26 | 123 | 109 |
| $BC_2F_6$ | 28 | 28 | 43 | 39 | 19 | 16 | 33 | 26 | 123 | 109 |

CM42: Chuanmai 42; JM22: Jimai 22; ZM9023: Zhengmai 9023; and XM26: Xinmai 26.

*3.2. Evaluation of Disease Resistance*

In the crop seasons of 2019–2020 and 2020–2021 in Mianyang and 2020–2021 in Yangling, the male parent, PI 660061, was highly resistant with IT 1–3, DS < 20%, and the recurrent parents CM42, JM22, ZM9023 and XM26 had IT 7–8, DS 60–80% with abundant uredinia (Figure 2). The majority of the 123 backcross progenies had IT values of 2–3 and DS values of 5–20% in the three environments (Figure 3). A total of 26 lines of $BC_2F_5$ and $BC_2F_6$ generations from CM42///CM42//CM42/PI660061 were resistant, while only 2 lines were intermediately resistant (Table 3). The number of resistant, intermediate and susceptible lines of JM22///JM22//JM22/PI660061 were 39, 2 and 2, respectively. The cross XM26///XM26//XM26/PI660061 produced 28 resistant lines, 4 intermediate lines and 1 susceptible line. The cross of ZM9023///ZM9023//ZM9023/PI660061 resulted in 17 resistant lines, 1 intermediate line and 1 susceptible line. Lines carrying *Yr59* had low IT (<4) and DS (<40) (Figure 4). The *t*-test results also prove the point (Figure 5). All these results indicated that introgression *Yr59* into elite wheat cultivars could effectively improve resistance to stripe rust. Additionally, lines possessing both *Yr59* and *Yr26* or both *Yr59* and *YrJ22* had mostly had higher levels of stripe rust resistance than those containing only *Yr59* (Table 4).

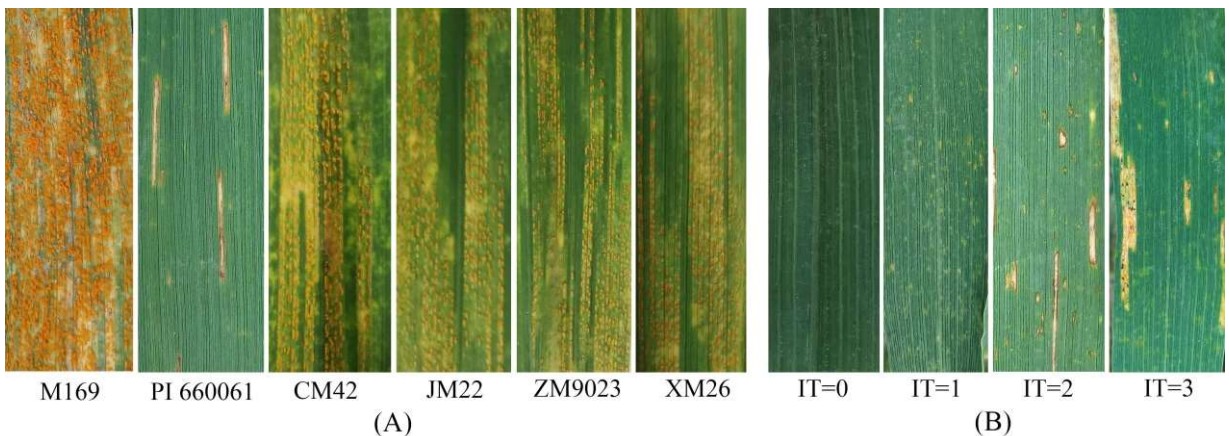

**Figure 2.** Adult-plant stripe rust reactions on leaves of five parents and susceptible control M169 (**A**); phenotype of sixteen final selected posterities, IT = 0 (including $BC_2F_6$-73), IT = 1 (including $BC_2F_6$-21, $BC_2F_6$-86, $BC_2F_6$-97), IT = 2 (including $BC_2F_6$-28, $BC_2F_6$-48, $BC_2F_6$-59, $BC_2F_6$-65, $BC_2F_6$-84, $BC_2F_6$-85, $BC_2F_6$-106, $BC_2F_6$-107), IT = 3 (including $BC_2F_6$-33, $BC_2F_6$-40, $BC_2F_6$-61, $BC_2F_6$-96) (**B**).

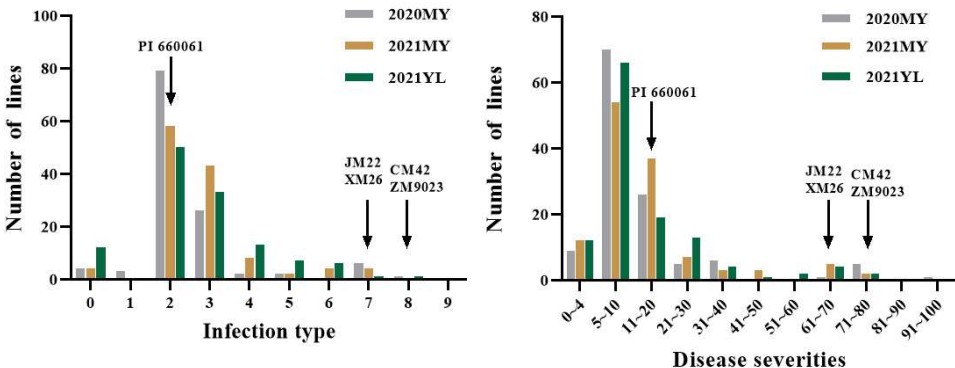

**Figure 3.** Distributions of mean infection type (IT) and disease severity (DS) for 123 Introgression Lines (ILs) from the four crosses of CM42///CM42//CM42/PI660061, JM22///JM22//JM22/PI660061, ZM9023///ZM9023//ZM9023/PI660061 and XM26///XM26//XM26/PI660061 at Yangling (YL) and Mianyang (MY) in 2019–2021. Arrows indicate the scores of the parental lines.

**Table 3.** Numbers of introgression lines of the four crosses classified in the resistant, intermediate and susceptible categories in the light of mean infection types (IT) over two years and three locations.

| Cross | Resistant (IT = 0–3) | Intermediate (IT = 4–6) | Susceptible (IT = 7–9) |
|---|---|---|---|
| CM42/PI660061 | 26 | 2 | 0 |
| JM22/PI660061 | 39 | 2 | 2 |
| ZM9023/PI660061 | 17 | 1 | 1 |
| XM26/PI660061 | 28 | 4 | 1 |

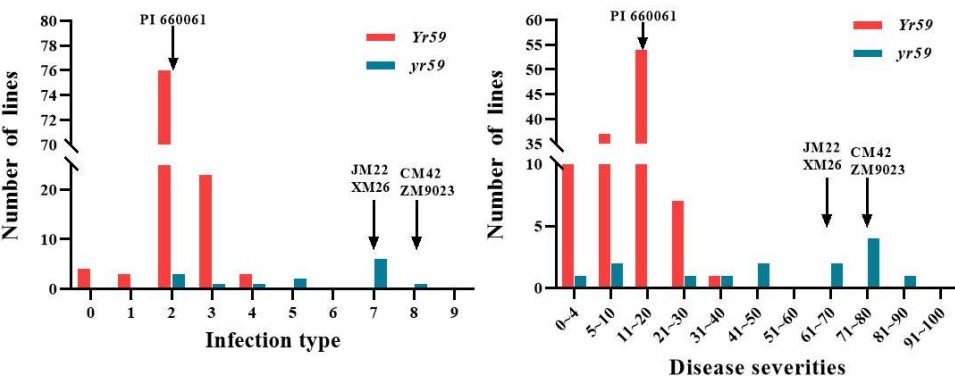

**Figure 4.** Comparison of 123 ILs for mean final infection type and disease severity about the presence (*Yr59*) and absence (*yr59*) of the stripe rust resistance gene *Yr59*.

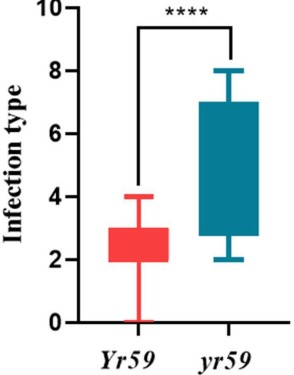

**Figure 5.** Infection type scores of *Yr59* and *yr59* lines from four populations. "*Yr59*" represents the presence of *Yr59*; "*yr59*" represents the absence of *Yr59*. *P*-value was generated by the *t*-test. **** Significantly different at $p < 0.05$ (n.s., no significance at $p > 0.05$).

**Table 4.** *Yr* genes detected by molecular markers, infection types (IT) and disease severity (DS) of stripe rust and agronomic traits of plant height (PH), number of spikes (NS), grain number per spike (GNS), spike length (SL) and thousand-grain weight (TWG) of parents and progeny lines.

| Line | Parent/Crosses | Stripe Rust | | | Agronomic Trait | | | | |
|---|---|---|---|---|---|---|---|---|---|
| | | Gene | IT | DS (%) | PH (cm) | NS | GNS | SL (cm) | TGW (g) |
| M169 | Susc. check | No | 9 | 100 | 116.0 | 5.0 | 39.0 | 10.3 | 38.1 |
| PI 660061 | Res. donor | *Yr59* | 2 | 20 | 125.1 | 8.0 | 45.0 | 8.7 | 39.5 |
| CM42 | Elite parent | *Yr26* | 6 | 75 | 91.3 | 5.0 | 40.0 | 7.1 | 42.8 |
| JM22 | Elite parent | *YrJ22* | 5 | 60 | 82.6 | 4.0 | 44.0 | 9.3 | 43.5 |
| ZM9023 | Elite parent | Unknown | 5 | 70 | 86.4 | 4.0 | 41.0 | 7.1 | 42.4 |
| XM26 | Elite parent | Unknown | 5 | 60 | 83.2 | 5.0 | 39.0 | 8.1 | 48.2 |
| $BC_2F_6$-7 | JM22/PI660061 | *Yr59+YrJ22* | 1 | 5 | 92.1 | 13.0 | 49.0 | 9.6 | 42.2 |
| $BC_2F_6$-8 | JM22/PI660061 | *Yr59+YrJ22* | 2 | 8 | 89.3 | 7.0 | 49.0 | 9.8 | 41.8 |
| $BC_2F_6$-21 * | JM22/PI660061 | *Yr59* | 1 | 7 | 93.5 | 9.0 | 55.0 | 9.0 | 46.6 |
| $BC_2F_6$-27 | JM22/PI660061 | *Yr59* | 3 | 20 | 95.3 | 10.0 | 47.0 | 11.1 | 42.3 |
| $BC_2F_6$-28 * | JM22/PI660061 | *Yr59* | 2 | 13 | 97.2 | 7.0 | 58.0 | 9.4 | 43.1 |
| $BC_2F_6$-33 * | JM22/PI660061 | *Yr59+YrJ22* | 3 | 5 | 88.1 | 9.0 | 59.0 | 10.0 | 44.7 |
| $BC_2F_6$-40 * | JM22/PI660061 | *Yr59+YrJ22* | 3 | 15 | 94.8 | 7.0 | 56.0 | 10.2 | 42.0 |
| $BC_2F_6$-48 * | CM42/PI660061 | *Yr59+Yr26* | 2 | 15 | 95.6 | 8.0 | 56.0 | 10.5 | 43.6 |
| $BC_2F_6$-59 * | CM42/PI660061 | *Yr59* | 2 | 8 | 97.8 | 7.0 | 56.0 | 10.1 | 47.5 |
| $BC_2F_6$-61 * | CM42/PI660061 | *Yr59+Yr26* | 3 | 18 | 99.7 | 7.0 | 61.0 | 10.0 | 46.0 |
| $BC_2F_6$-65 * | CM42/PI660061 | *Yr59* | 2 | 13 | 99.3 | 5.0 | 56.0 | 9.0 | 52.7 |
| $BC_2F_6$-69 | CM42/PI660061 | *Yr59+Yr26* | 3 | 17 | 90.6 | 7.0 | 51.0 | 9.4 | 43.8 |
| $BC_2F_6$-72 | ZM9023/PI660061 | *Yr59* | 2 | 8 | 85.1 | 6.0 | 43.0 | 8.9 | 44.2 |
| $BC_2F_6$-73 * | ZM9023/PI660061 | *Yr59* | 0 | 0 | 88.7 | 6.0 | 46.0 | 8.3 | 57.6 |
| $BC_2F_6$-78 | ZM9023/PI660061 | *Yr59* | 2 | 13 | 97.0 | 8.0 | 46.0 | 8.4 | 40.3 |
| $BC_2F_6$-84 * | ZM9023/PI660061 | *Yr59* | 2 | 4 | 97.7 | 8.0 | 45.0 | 10.2 | 58.1 |
| $BC_2F_6$-85 * | ZM9023/PI660061 | *Yr59* | 2 | 17 | 86.1 | 7.0 | 45.0 | 9.3 | 50.0 |
| $BC_2F_6$-86 * | ZM9023/PI660061 | *Yr59* | 1 | 5 | 96.2 | 7.0 | 42.0 | 10.3 | 57.8 |
| $BC_2F_6$-96 * | XM26/PI660061 | *Yr59* | 3 | 12 | 82.8 | 7.0 | 51.0 | 9.2 | 49.4 |
| $BC_2F_6$-97 * | XM26/PI660061 | *Yr59* | 2 | 5 | 99.4 | 7.0 | 43.0 | 10.2 | 53.7 |
| $BC_2F_6$-98 | XM26/PI660061 | *Yr59* | 2 | 4 | 87.1 | 5.0 | 43.0 | 8.7 | 54.8 |
| $BC_2F_6$-102 | XM26/PI660061 | *Yr59* | 3 | 15 | 96.8 | 6.0 | 56.0 | 8.5 | 44.1 |
| $BC_2F_6$-106 * | XM26/PI660061 | *Yr59* | 3 | 20 | 96.5 | 6.0 | 61.0 | 10.5 | 44.1 |
| $BC_2F_6$-107 * | XM26/PI660061 | *Yr59* | 3 | 13 | 88.6 | 7.0 | 60.0 | 9.3 | 43.0 |
| $BC_2F_6$-116 | XM26/PI660061 | *Yr59* | 3 | 12 | 94.8 | 6.0 | 58.0 | 9.2 | 42.6 |

* The lines were eventually selected to be released as new germplasm resource.

### 3.3. Evaluation of Agronomic Traits

Various agronomic traits (including PH, NS, GNS, SL and TWG) of the parents and 123 $BC_2F_5$ or $BC_2F_6$ lines were assessed during the 2019–2020 and 2020–2021 growing season in Mianyang and Yangling. The mean PH of five parents: PI 660061, CM42, JM22, ZM9023 and XM26 were 125.1, 91.3, 82.6, 86.4, 83.2, respectively, whereas the selected progeny lines had 80.0–110.0 cm (Figure 6), which were desirable for growing in the regions of the elite cultivar parents from. The mean NS values of the five parents were 8.0, 5.0, 4.0, 4.0 and 5.0, respectively, and the NS values in the selected progeny lines exceeded that of their female parents, mostly ranged from 6.0 to 10.0. The mean SL value of each parent was 8.7 cm of PI660061, 7.1 cm of CM42, 9.3 cm of JM22, 7.1 cm of ZM9023 and 8.1 cm of XM26. Their selected plants had SL values mostly between 8.0 and 11.0 cm. The mean GNS values of the five parents were 45.0, 40.0, 44.0, 41.0 and 39.0, while the selected progeny lines had 46.0–60.0 grains per spike, significantly higher than their donor parent and recurrent parents. The mean TWG values of the five parents were 39.5, 42.8, 43.5, 42.4 and 48.2 g, respectively while the mean TWG values of their selected progeny lines majorly ranged from 40.0 to 50.0 g. The TWG values of the selected progeny lines from cross JM22///JM22//JM22/PI660061 ranged mainly from 30.0 to 50.0 g (Figure 6). These

important agronomic traits had been improved to varying degrees for reaching the goal of releasing new cultivars in the various wheat growing regions.

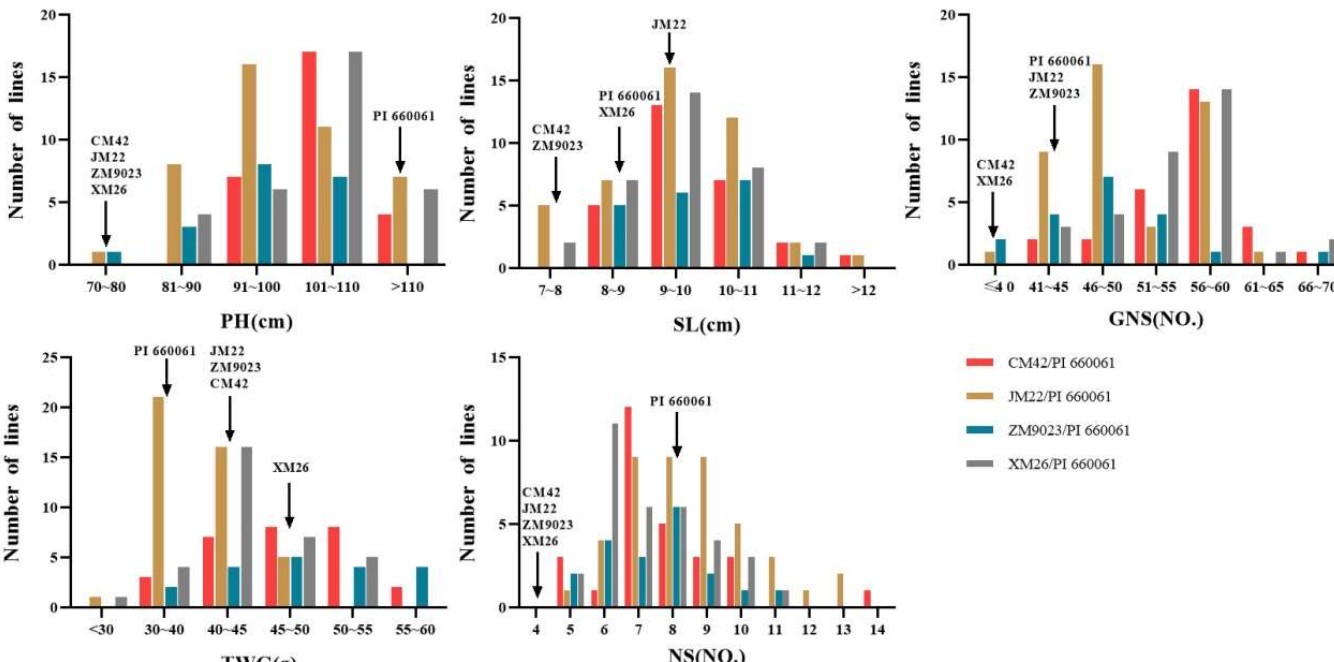

**Figure 6.** Frequency distributions of agronomic traits for the 123 ILs from the four populations. plant height (PH); spike length (SL); grain number per spike (GNS); thousand-grain weight (TGW); number of spikes (NS).

As elite wheat cultivars with excellent agronomic traits, the four recurrent parents were used as the reference standards for selection of the final lines with *Yr59*. The 25 selected lines had PH between 80.0 and 100.0 cm for lodging resistance and easy harvesting; NS and GNS equal to or more than 5.0 and 40.0, respectively; SL over 8.0 cm; and TGW of 40.0 g or greater (Table 4). Of the 25 lines, 5 were from the cross of CM42///CM42//CM42/PI660061, 7 from JM22///JM22//JM22/PI660061, 6 from ZM9023///ZM9023//ZM9023/ZM9023 and 7 from XM26///XM26//XM26/PI660061. These lines had IT 0 to 3 and DS less than 20%. Further narrowed selection by screening suitable tiller numbers to prevent lodging and maximize wheat yield. Eventually, 16 lines were selected from the 25 lines, 4 from each of the 4 crosses. These lines were $BC_2F_6$-21, $BC_2F_6$-28, $BC_2F_6$-33 and $BC_2F_6$-40 (from JM22///JM22//JM22/PI660061); $BC_2F_6$-48, $BC_2F_6$-59, $BC_2F_6$-61 and $BC_2F_6$-65 (from CM42///CM42//CM42/PI660061); $BC_2F_6$-73, $BC_2F_6$-84, $BC_2F_6$-85 and $BC_2F_6$-86 (from ZM9023///ZM9023//ZM9023/PI660061); $BC_2F_6$-96, $BC_2F_6$-97, $BC_2F_6$-106 and $BC_2F_6$-107 (from XM26///XM26//XM26/PI660061). Among the 16 final lines, $BC_2F_6$-48 and $BC_2F_6$-61 had *Yr26*, and $BC_2F_6$-33 and $BC_2F_6$-40 had *YrJ22* for race-specific ASR, in addition to *Yr59* for non-race-specific HTAP resistance. The remaining 12 lines had *Yr59*. The specific results are shown in Table 4.

In summary, we successfully introduced the HTAP stripe rust resistance gene *Yr59* into four elite wheat cultivars from different wheat growing regions in China. Through MAS, 109 families with resistance gene *Yr59* were selected from 123 lines, and 16 lines were finally selected based on their desirable agronomic traits. These lines showed high resistance to stripe rust (IT ≤ 3; DS ≤ 20%); moderate plant height (80–100 cm); and yield-related traits including NS (5–9), SL (8.38–10.59) and GNS (42–61), and TGW were similar or superior to their elite parents, providing more options for wheat breeding.

## 4. Discussion

Stripe rust is prevalent in many wheat areas in China and has caused serious damage. One of the main reasons for the frequent epidemics is the widely grown single wheat cultivars with one or few race-specific resistance genes, which creates directional selection pressure on the pathogen to produce new predominant races, circumventing the resistance genes [64]. As a result, the wheat cultivars become susceptible. With the development of MAS technology, wheat cultivars containing different resistance genes have been developed. For example, pyramiding lines (consisting of 3–8 *Yr* genes) were constructed by marker assistant selection for better stripe rust resistance [65]. Multiple genes/QTL, such as *Yr70/Lr76 + Lr37/Yr17/Sr38, Gpc-B1/Yr36 + QPhs.ccsu-3A.1 + QGw.ccsu-1A.3 + Lr24/Sr24 + Glu-A1-1/Glu-A1-2*, were successfully pyramided into wheat cultivars to improve resistance to stripe rust and other rusts as well as grain quality [66]. Wheat cultivar Guinong 19 was developed from the cross of Zhongyan 96-3 × Guinong 21 through MAS for powdery mildew resistance gene *Pm21* [67].

Race CYR34 of Pst, which is virulent to resistance genes *Yr26* (=*Yr24*) and *Yr10*, has become predominant in recent years [68], resulting in many wheat cultivars with *Yr26* becoming susceptible, such as Guinong 22 and Chuanmai 42, both with *Yr26*, as well as Zhengmai 9023 with unknown stripe rust resistance genes. Through our disease resistance assessments in the field over the years, PI 660061 has still maintained resistance to CYR34 and many other races, as its HTAP resistance controlled by *Yr59* is non-race-specific [36]. As the gene was recently identified from an Iraq landrace (PI 178759), its use in breeding programs has just started. As PI 660061 was developed as an improved germplasm from PI 178759 for *Yr59* [56], PI 660061 was selected as the donor parent to cross with four high yield wheat cultivars, CM42, JM22, ZM9023 and XM26 from different wheat growing regions of China. Four markers, *Xbarc32*, *Xwgp5175*, *Xwmc557* and *Xcfa2040*, linked with *Yr59* were used to detect the gene in the progeny populations. Given three of the four are SSR markers, recombination between markers and gene may occur, which might affect the results of marker detection. Thus, we combined marker selection with field stripe rust screening to ensure as much as possible that *Yr59* was successfully introduced into the backgrounds of the four recipient parents. Finally, through the screening of agronomic characters, 16 lines with *Yr59* in combinations with multiple desirable agronomic traits were selected.

According to the reports, CM42 and JM22 have race-specific ASR genes *Yr26* and *YrJ22*, respectively [48–50]. In our results, almost all lines carrying *Yr59* showed high level of resistance with IT scores lower than 3 and DS less than 20%. Some lines having *Yr59* together with *Yr26* or *YrJ22* exhibited stronger resistance than lines containing only *Yr59* from crosses CM42///CM42//CM42/PI660061 and JM22/PI660061//JM22///JM22. This may because of the combination of effects from both genes. Surprisedly, the enhancing effect may also happen in the crosses of ZM9023/PI660061//ZM9023///ZM9023 and XM26///XM26//XM26/PI660061. There were several lines without *Yr59* in ZM9023/// ZM9023//ZM9023/PI660061 and XM26///XM26//XM26/PI660061 that displayed some levels of resistance. As ZM9023 and XM26 were less susceptible than the susceptible check (Table 4), they should have unknown resistance genes. The combinations of the unknown resistance genes with *Yr59* should enhance resistance. A recent study reported that ZM9023 contains *Yr29* and *Yr30* for adult-plant resistance to stripe rust as well as *Lr46/Sr59/Pm39* and *Lr27/Sr2* for resistance to leaf rust/stem rust or powdery mildew linked to or pleiotropic with the stripe rust resistance genes, respectively [69].

Through our multi-year field tests, we showed that PI 660061 was easy lodging due to high plant height and excessive tillers. In the present study, PI 660061 was crossed with four superior cultivars, and this unfavorable trait was significantly improved. Yield is another important trait in wheat production, which was reflected partly by the number of grains per spike and 1000-grain weight. Pyramided *Yr59* into the backgrounds of these four cultivars helped to improve the stripe rust resistance and maintained the similar or even improved the traits of grains per spike and number of spikes compared with their recurrent parents. In the backcross of JM22///JM22//JM22/PI660061, a half of the progeny lines

had thousand-grain weight less than their recurrent parent, but this may be compensated by the improved spike grain number and number of spikes to a large extent.

Generally, MABC requires both foreground selection and background selection. In this study, we did not carry out marker detection and comprehensive evaluation selection in every generation. Instead, marker-assisted selection was carried out in the $F_1$, $BC_1F_1$ and $BC_2F_2$ generations, and the selected $BC_2F_2$ plants were subsequently advanced through selfing to the $BC_2F_4$ generation for comprehensive evaluation and selection of agronomic traits and stripe rust resistance. To preserve all potential genotypes and reduce the loss of desirable heterozygous plants, bulk harvest was performed before the $BC_2F_4$ generation [70]. Final selection was done based on phenotyping for the resistance and agronomic traits. Because Mianyang is a hot spot of stripe rust, it provides conditions for reliable identification of stripe rust resistance. Thus, we screened for stripe rust resistance in the $BC_2F_4$ generation. Through preliminary screening of stripe rust resistance and agronomic traits of $BC_2F_4$ in the field, we found that the proportion of lines with stripe rust resistance was significantly increased, saving the cost and labor for the next step of marker-assisted selection. This procedure reduces the working intensity and ensures the correctness of the selection. Nevertheless, the lines we selected in this study were stable. Marker detection for targeting stripe rust resistance genes and phenotype evaluation of stripe rust resistance and agronomic traits were performed in both the $BC_2F_5$ (Mianyang) and $BC_2F_6$ (Mianyang and Yangling) generations. The consistent results in multiple environments prove the reliability of the selection results.

## 5. Conclusions

The emergence and prevalence of new Pst races may have destructive impacts on wheat production, which requires timely excavation and utilization of new disease-resistance genes, even better genes for non-race-specific, adequate and durable resistance. In the current study, we successfully transferred *Yr59* for an adequate level of HTAP resistance, a type of stripe rust resistance that has been demonstrated to be non-race-specific and durable [31,71,72], into four Chinese elite wheat cultivars and selected 16 lines with strong stripe rust resistance and desirable agronomic traits through MABC. These pyramided lines provide genetic resources for breeding new wheat cultivars with durable resistance to stripe rust and may be released as new cultivars to certain wheat production regions after evaluation for yield, quality, adaptation and resistance to other major diseases and pests in various regions.

**Author Contributions:** M.Z. and X.Z. detected the gene, collected samples, analyzed the data and prepared the first draft of the manuscript. M.Z. and T.F. contributed to field investigation and samples collection. X.Z. and X.C. contributed to the crosses and revised the draft. X.L., J.F. and S.Y. contributed to the selection target lines and evaluated the populations. X.Z., X.C. and Z.K. conceived the project and generated the final version of the manuscript. All authors have read and agreed to the published version of the manuscript.

**Funding:** This work was financially supported by the Key Research and Development Program of International Science and Technology Innovation Cooperation of Science and Technology Department of Sichuan Province, China (No. 22GJHZ0288), and was partially funded by the National Natural Science Foundation of China (No. 32101707), PhD Foundation of Southwest University of Science and Technology (No. 18zx7159), Breakthrough in Wheat Breeding Material and Method Innovation and New Variety Breeding (Breeding Research Project, 2021YFYZ0002), Longshan Academic Talent Research Support Program of SWUST (No. 17LZX5) and PhD Foundation of Southwest University of Science and Technology (No. 16zx7162).

**Institutional Review Board Statement:** Not applicable.

**Informed Consent Statement:** Not applicable.

**Data Availability Statement:** The datasets analyzed during the current study are available from the corresponding author on reasonable request.

**Acknowledgments:** The authors acknowledge the Crop Molecular Breeding Platform of Sichuan Province for providing necessary infrastructure and research facilities for carrying out this work and Biotechnology and Nuclear Technology Research Institute, Sichuan Academy of Agricultural Sciences, for assistance with this research.

**Conflicts of Interest:** The authors declare no conflict of interest.

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
