# Peer review of "Combination of Marker-Assisted Backcross Selection of Yr59 and Phenotypic Selection to Improve Stripe Rust Resistance and Agronomic Performance in Four Elite Wheat Cultivars"

_agronomy, doi:10.3390/agronomy12020497_

Round 1

Reviewer 1 Report

Сomments:

  1. The title does not fully match the content of the article. The text presents data on obtaining new breeding lines with resistance genes and agronomic traits, and it is not entirely correct to talk about gene introgression into the studied varieties.
  2. The plant material and methods are poorly described. The text does not contain information about lifestyle of parent cultivars CM42, ZM9023 and line PI 178759.  

  3. Information about the places of cultivation and phenotyping of different generations requires clarification. There are inconsistencies in the text. Section 2.1 line 141-142: «field phenotypic tests before 2017 were conducted in Yangling (34.292N, 108.077E) of Shanxi province, and the 2018-2021 tests were mainly conducted in Mianyang (31.682N, 104.663E) of Sichuan province». However, in section 2.2. line 149-152 find: «The BC2F5 seeds were planted and their plants were evaluated for stripe rust resistance and agronomic traits in Mianyang during the 2019-2020 growth season, and the BC2F6 line 274: were planted and evaluated in the experimental fields of both Mianyang and Yangling during the 2020-2021 season».

  4. Statement (section 3.2 lines 287-289): «Additionally, lines possessing both Yr59 and Yr26 or both Yr59 and YrJ22 had mostly had higher levels of stripe rust resistance than those containing only Yr59» is not correct, does not correspond to the data in Table 4. (Yr59 and+ Yr26 have IT 1-3, DS 5- 15, Yr59 - 1-3, 7-20; Yr59 + YrJ22 - IT 2-3, DS - 15-18, Yr59 IT 2, DS - 8-13).

Reviewer 2 Report

The manuscript entitled “Introgression of Genes of Yr59 into Four Chinese Popular Wheat Cultivars Using Marker-Assisted Backcross Selection” authored by Zhang et al. has been written well. However, a number of minor improvements are required before accepting this paper.

  1. The abstract has 250 words, but according to the journal, the abstract should be a total of about 200 words maximum.
  2. Scientific names and their short form like “Puccinia striiformis sp. tritici (Pst)” and symbol for genes like “Yr59” generally should be italicized.
  3. Line 42: (Pst), the bracket need to be deleted.
  4. Line 50: “improvement” ---> "the improvement"
  5. Line 205: “dd H2O” ---> “double distilled water (ddH2O)”
  6. Line 264: Table 2 needs more space between the column, especially among the four crosses and total. Otherwise, it can confuse the reader at the first sight. Please follow the “Table 2 of Agronomy template, Weblink; https://www.mdpi.com/files/word-templates/agronomy-template.dot”, so that you will get more space among the column. You can do this for Table 4 also.
  7. Line 295-298: It is better to align left for the title of figure 3.
  8. Line 295, 304, 334: Please define “ILs” as “Introgression Lines” at least the first time in line 295.
  9. Line 394: Citation missing.
  10. Line 401: Space needed before There.
  11. Line 409: Citation missing.

Author Response

Please see the attachment。
